# DROSHA-Dependent AIM2 Inflammasome Activation Contributes to Lung Inflammation during Idiopathic Pulmonary Fibrosis

**DOI:** 10.3390/cells8080938

**Published:** 2019-08-20

**Authors:** Soo Jung Cho, Kyoung Sook Hong, Ji Hun Jeong, Mihye Lee, Augustine M. K. Choi, Heather W. Stout-Delgado, Jong-Seok Moon

**Affiliations:** 1Joan and Sanford I. Weill Department of Medicine, Weill Cornell Medical College, New York-Presbyterian Hospital, New York, NY 10011, USA; 2Division of Pulmonary and Critical Care Medicine, Weill Cornell Medical College, New York, NY 10011, USA; 3Department of Surgery and Critical Care Medicine, Ewha Womans University College of Medicine, Seoul 05505, Korea; 4Department of Integrated Biomedical Science, Soonchunhyang Institute of Medi-bio Science (SIMS), Soonchunhyang University, Cheonan-si 31151, Korea

**Keywords:** DROSHA, miRNA, AIM2 inflammasome, idiopathic pulmonary fibrosis

## Abstract

Idiopathic pulmonary fibrosis (IPF) has been linked to chronic lung inflammation. Drosha ribonuclease III (DROSHA), a class 2 ribonuclease III enzyme, plays a key role in microRNA (miRNA) biogenesis. However, the mechanisms by which DROSHA affects the lung inflammation during idiopathic pulmonary fibrosis (IPF) remain unclear. Here, we demonstrate that DROSHA regulates the absent in melanoma 2 (AIM2) inflammasome activation during idiopathic pulmonary fibrosis (IPF). Both DROSHA and AIM2 protein expression were elevated in alveolar macrophages of patients with IPF. We also found that DROSHA and AIM2 protein expression were increased in alveolar macrophages of lung tissues in a mouse model of bleomycin-induced pulmonary fibrosis. DROSHA deficiency suppressed AIM2 inflammasome-dependent caspase-1 activation and interleukin (IL)-1β and IL-18 secretion in primary mouse alveolar macrophages and bone marrow-derived macrophages (BMDMs). Transduction of microRNA (miRNA) increased the formation of the adaptor apoptosis-associated speck-like protein containing a caspase recruitment domain (ASC) specks, which is required for AIM2 inflammasome activation in BMDMs. Our results suggest that DROSHA promotes AIM2 inflammasome activation-dependent lung inflammation during IPF.

## 1. Introduction

Pulmonary fibrotic disorders include idiopathic pulmonary fibrosis (IPF), hypersensitivity pneumonitis, sarcoidosis, radiation-induced fibrosis, and the interstitial fibrosis associated with collagen vascular diseases such as rheumatoid arthritis and systemic lupus erythematous [1]. IPF is the most common of these interstitial lung diseases and has an incidence in northern America estimated at between 6 and 18 per 100,000 individuals [2]. It is characterized by focal areas of chronic inflammation, aberrant immune response, epithelial cell injury, proliferation of mesenchymal cells, and dysregulated wound healing, ultimately leading to progressive respiratory failure and death [3,4]. Although several risk factors for pulmonary fibrosis including cigarette smoking, bacterial and viral infection, and exposure to environmental particles have been identified, the etiology of IPF remains unknown [5,6].

Among the exacerbated inflammatory response in IPF, alveolar macrophages, long-lived resident innate immune cells of the airways, are key effectors of recognition, initiation, and resolution of the host defense against microbes [7,8,9]. Inflammasomes, which are large cytosolic multi-protein complexes, are activated by assembly in response to pathogen-associated molecular patterns (PAMPs) or damage-associated molecular patterns (DAMPs) and lead to caspase-1-mediated inflammatory responses including cleavage and secretion of the pro-inflammatory cytokines interleukin (IL)-1β and IL-18 [10,11]. Inflammasomes consist of sensor proteins, such as nucleotide oligomerization domain (NOD)-, leucine-rich repeat (LRR)- and pyrin domain-containing 1 (NLRP1), NLRP3, and NLRC4, or pyrin and HIN domain-containing protein (PYHIN), family members absent in melanoma 2 (AIM2), caspase-1, and the adaptor apoptosis-associated speck-like protein containing a caspase recruitment domain (ASC), which is an adaptor protein encoded by PYRIN-PAAD-DAPIN domain (PYD) and a C-terminal caspase-recruitment domain (CARD) (PYCARD) that is a common protein in all inflammasomes [12,13,14]. In the formation of inflammasome, ASC interacts with both the upstream inflammasome sensor protein and caspase-1, and this interaction triggers the assembly of ASC as a large protein speck for inflammasome activation [15,16].

Absent-in-melanoma-2 (AIM2), a prototypical member of the AIM2-like-receptor (ALR) family, is a receptor for cytosolic double stranded DNA (dsDNA) from PAMPs and self-DNA from DAMPs [17,18]. Binding of dsDNA to AIM2 leads to the activation of AIM2 inflammasome complex by recruitment of ASC via PYD–PYD interactions and caspase-1 via CARD–CARD interactions [19]. The formation of AIM2 inflammasome promotes the increase of caspase-1 p10 fragments as an active form. Activated caspase-1 p10 fragments proteolytically produce activated IL-1β and IL-18 from pro-IL-1β and pro-IL-18 [20,21]. We have previously reported that the activation of inflammasome is linked to inflammatory diseases such as sepsis [22]. Although the activation of inflammasome was introduced in pulmonary diseases, the activation of AIM2 inflammasome in IPF pathogenesis remains unclear.

MicroRNAs (miRNAs) are short non-coding RNA molecules, usually composed of 18–25 nucleotides, whose biogenesis begins with the synthesis of primary miRNA (pri-miRNA) by the actions of RNA polymerase III and II [23,24]. The stem-loop structure of pri-miRNA is cleaved by the microprocessor complex composed of DROSHA and its cofactor DiGeorge syndrome critical region 8 (DGCR8) [23,24]. The miRNAs are expressed in a wide variety of organs and cells and are known to regulate inflammatory actions by gene expression at the posttranscriptional level by degrading their target mRNAs and/or inhibiting their translation [25,26]. The changes in miRNAs have been reported in IPF patients [27,28,29,30]. The specific miRNAs are associated with regulation of differentiation, proliferation, cell–cell interaction and fibroblast activation in the pulmonary fibrosis [31,32,33,34,35,36,37,38]. However, the role of DROSHA in pulmonary inflammation in IPF pathogenesis remains unclear.

Here, we show that both DROSHA and AIM2 protein expression are significantly elevated in alveolar macrophages of patients with IPF and a mouse model of bleomycin-induced pulmonary fibrosis. Moreover, we demonstrate that deficiency of DROSHA significantly suppresses the AIM2 inflammasome activation in alveolar macrophages. Furthermore, the transduction of miRNA significantly promotes the formation of ASC specks which is required for AIM2 inflammasome activation. These results suggest that DROSHA promotes AIM2 inflammasome-dependent lung inflammation during IPF pathogenesis.

## 2. Materials and Methods

### 2.1. Human Subjects

Human subjects study was conducted in accordance with the Helsinki Declaration. The Brigham and Women’s Hospital and Weill Cornell Medicine institutional review board approved all experimental procedures involving use of human samples (protocol #: 1405015116A003). Fresh frozen human lung tissues were collected from IPF patients undergoing lung transplantation or from failed donor controls. A pulmonary, radiology and pathology multi-disciplinary expert group ascertained the diagnosis of idiopathic pulmonary fibrosis using american thoracic society (ATS) guidelines [39].

### 2.2. Animal Studies

All mouse experimental protocols were approved by the Institutional Animal Care and Use Committee of Soonchunhyang University (protocol #: SCH18-0032) and Weill Cornell Medicine (protocol #: 2013-0108). C57BL/6 mice (male, 2 months old) were purchased from the National Institute on Aging Mouse Colony. Mice were exposed to phosphate-buffered saline or bleomycin (0.01 mg/mouse) via oropharyngeal aspiration. All animal experimental protocols were approved by the institutional animal care and use committee at Weill Cornell Medicine (protocol #: 2013-0108).

### 2.3. Reagents and Antibodies

Ultrapure lipopolysaccharide (LPS) (tlrl-3pelps), flagellin (*Salmonella typhimurium*) (tlrl-stfla), and MDP (tlrl-mdp) were from Invivogen (San Diego, CA, USA). The poly(dA:dT) (P0883) and ATP (A2383) were from Sigma-Aldrich (St. Louis, MO, USA). Recombinant Mouse transforming growth factor (TGF)-beta 1 Protein (7666-MB-005) was from R&D systems (Minneapolis, MN, USA)) The following antibodies were used: monoclonal rabbit anti-DROSHA for human and mouse DROSHA (1:1000) (#3410, Cell signaling (Danvers, MA, USA)); monoclonal rabbit anti-DGCR8 for human and mouse DGCR8 (1:1000) (ab191875, Abcam (Cambridge, MA, USA)); monoclonal rat anti-F4/80 for human and mouse F4/80 (1:1000) (MCA497RT, Bio-Rad Laboratories (Hercules, CA, USA)); monoclonal mouse anti-CD68 for humans (ab955, Abcam (Cambridge, MA, USA)); polyclonal rabbit anti-AIM2 for human and mouse AIM2 (1:1000) (ab93015, Abcam (Cambridge, MA, USA)); monoclonal mouse anti-AIM2 for humans (1:1000) (ab204995, Abcam (Cambridge, MA, USA)); polyclonal rabbit anti-caspase-1 for mouse caspase-1 (1:1000) (SC-514, SantaCruz Biotechnology (Dallas, TX, USA)); polyclonal goat anti-IL-1β for mouse IL-1β (1:1000) (AF-401-NA, R&D Systems (Minneapolis, MN, USA)); polyclonal rabbit anti-ASC for mouse ASC (1:1000) (ADI-905-173-100, Enzo Lifesciences (Farmingdale, NY, USA)); Monoclonal Anti-Actin, α-Smooth Muscle (1:1000) (A2547, Sigma-Aldrich (St. Louis, MO, USA)); Anti-Collagen I antibody (1:1000) (ab21286, Abcam (Cambridge, MA, USA)); Anti-Fibronectin antibody (1:1000) (ab2413, Abcam (Cambridge, MA, USA)); Goat anti-Mouse IgG1 Cross-Adsorbed Secondary Antibody, Alexa Fluor 488 (A-21121, Thermo Fisher Scientific (Waltham, MA, USA)); Goat anti-Rabbit IgG (H + L) Highly Cross-Adsorbed Secondary Antibody, Alexa Fluor 555 (A-21429, Thermo Fisher Scientific (Waltham, MA, USA)); Goat anti-Rat IgG (H + L) Cross-Adsorbed Secondary Antibody, Alexa Fluor 594 (A-11007, Thermo Fisher Scientific (Waltham, MA, USA)), and monoclonal mouse anti-β-actin (1:5000) (A5316, Sigma-Aldrich).

### 2.4. Cell Culture

Primary mouse alveolar macrophages were obtained by bronchoalveolar lavage (BAL) from wild-type (WT) mice (males, 10–12 weeks old) and were cultured in DMEM medium (Invitrogen (Waltham, MA, USA)) containing 10% (*v*/*v*) heat-inactivated fetal bovine serum (FBS), 100 units/mL penicillin, and 100 mg/mL streptomycin. For primary mouse bone marrow-derived macrophages (BMDMs), bone marrow collected from wild type (WT) mouse (male, 8–10 weeks old) femurs and tibias was plated on sterile Petri dishes and incubated for 7 days in DMEM medium (Invitrogen (Waltham, MA, USA)) containing 10% (*v*/*v*) heat-inactivated fetal bovine serum (FBS), 100 units/mL penicillin, 100 mg/mL streptomycin, and 25% (*v*/*v*) conditioned medium from mouse L929 fibroblasts (CCL-1, ATCC (Manassas, VA, USA)). For the AIM2 inflammasome activation, LPS-primed wild-type (WT) bone marrow-derived macrophages (BMDMs) were transfected with poly(dA:dT) (1 mg/mL, 3 h) (Sigma-Aldrich) using Lipofectamine with Plus reagent (15338-100, Invitrogen), according to the manufacturer’s instructions. For the NLRP3 inflammasome activation, LPS-primed WT BMDMs were treated with ATP (2 mM, 0.5 h) (Sigma-Aldrich). For the NLRC4 inflammasome activation, LPS-primed WT BMDMs were transfected with flagellin (5 μg/mL, 6 h) (Invivogen) using Lipofectamine with Plus reagent (15338-100, Invitrogen), according to the manufacturer’s instructions. For the NLRP1 inflammasome activation, LPS-primed WT BMDMs were treated with MDP (5 μg/mL, 16 h) (Invivogen). For cytokine analysis, the cell supernatants were collected and analyzed for the levels of IL-1β, IL-18, and TNF-α using a ELISA kit. Primary mouse lung fibroblasts were isolated and cultured as described previously [40]. Primary mouse lung fibroblasts (2 × 10^5^ cells in 6-well cell culture plates) were treated with TGF-β (20 ng/mL, 24 h).

### 2.5. Transduction of Drosha gRNA/Cas9 pPlasmid or Drosha Small Interfering RNA (siRNA)

For stable knockdown of mouse *Drosha*, *Drosha*-targeting gRNA (TACGTGGTAAGTGGTATTCT) in pCas-Guide CRISPR plasmid (Cat. No. KN304817, OriGene Technologies (Rockville, MD, USA)) was used. We used CRISPR ribonucleoprotein (RNP) system (GeneScript (Piscataway, NJ, USA)). To prepare RNA oligo, 10 μL DROSHA-targeting single guide RNA (sgRNA) Oligo (100 μM) were incubated at 95 °C for 5 min with anneal components (Nuclease-Free Water (22 μL) and Annealing Buffer (5X) (8 μL). This was put in 60 °C water and left to cool to room temperature. To prepare and transduce ribonucleoprotein (RNP) complex, WT alveolar macrophages (2 × 10^5^ cells in 6-well cell culture plates) were seeded and transfected with RNP complex (Cas9 Nuclease (Z03386, GeneScript) 15 pmol and sgRNA oligos annealed 30 pmol) using Lipofectamine™ CRISPRMAX™ Cas9 Transfection Reagent (CMAX00003, Invitrogen) according to the manufacturer’s instructions. BMDMs were incubated for 48 h and stimulated LPS and poly(dA:dT) as described. For transient knockdown of mouse DROSHA, MISSION^®^ esiRNA esiRNA targeting mouse Rnasen (Cat. No. EMU046711, Sigma-Aldrich) was used. WT BMDMs (2 × 10^5^ cells in 6-well cell culture plates) were seeded and transfected with siRNA for mouse DROSHA or control siRNA for control using Lipofectamine with Plus reagent (15338-100, Invitrogen) according to the manufacturer’s instructions.

### 2.6. Synthetic miRNAs

For transient transduction of synthetic miRNAs, miRNA#1 (mmu-miR-223, UCUGGCCAUCUGCAGUGUCACGCUCCGUGUAUUUGACAAGCUGAGUUGGACACUCUGUGUGGUAGAGUGUCAGUUUGUCAAAUACCCCAAGUGUGGCUCAUGCCUAUCAG), miRNA#2 (mmu-miR-200b, GCCGUGGCCAUCUUACUGGGCAGCAUUGGAUAGUGUCUGAUCUCUAAUACUGCCUGGUAAUGAUGACGGC), miRNA#3 (mmu-miR-191, AGCGGGCAACGGAAUCCCAAAAGCAGCUGUUGUCUCCAGAGCAUUCCAGCUGCACUUGGAUUUCGUUCCCUGCU), miRNA#4 (mmu-miR-155, CUGUUAAUGCUAAUUGUGAUAGGGGUUUUGGCCUCUGACUGACUCCUACCUGUUAGCAUUAACAG) and miRNA#5 (mmu-miR-208a, UUCCUUUGACGGGUGAGCUUUUGGCCCGGGUUAUACCUGACACUCACGUAUAAGACGAGCAAAAAGCUUGUUGGUCAGAGGAG) were used. WT BMDMs (2 × 10^5^ cells in 6-well cell culture plates) were seeded and transfected with miRNAs or control using Lipofectamine with Plus reagent (15338-100, Invitrogen) according to the manufacturer’s instructions.

### 2.7. Immunoblot Analysis

The WT alveolar macrophages and BMDMs were harvested and lysed in NP40 Cell Lysis Buffer (FNN0021, Invitrogen). The lysates were centrifuged at 15,300× *g* for 10 min at 4 °C, and the supernatants were obtained. The protein concentrations of the supernatants were determined by applying the Bradford assay (500-0006, Bio-Rad Laboratories (Hercules, CA, USA)). Proteins were electrophoresed on NuPAGE 4–12% Bis-Tris gels (Invitrogen) and transferred to Protran nitrocellulose membranes (10600001, GE Healthcare Life Science (Pittsburgh, PA, USA)). The membranes were blocked in 5% (*w/v*) bovine serum albumin (BSA) (9048-46-8, Santa Cruz Biotechnology) in TBS-T (TBS (170-6435, Bio-Rad Laboratories) and 1% (*v*/*v*) Tween-20 (170-6531, Bio-Rad Laboratories)) for 30 min at 25 °C. The membranes were incubated with primary antibody diluted in 1% (*w/v*) BSA in TBS-T for 16 h at 4 °C and then with the horseradish peroxidase (HRP) conjugated secondary antibody (anti-rabbit IgG HRP (SC-2357, SantaCruz Biotechnology) (1:2500) and anti-mouse m-IgGκ BP-HRP (SC-516102, SantaCruz Biotechnology) (1:2500)) diluted in TBS-T for 30 min at 25 °C. The immunoreactive bands were detected using the SuperSignal West Pico Chemiluminescent Substrate (34078, Thermo Scientific (Waltham, MA, USA)).

### 2.8. Immunohistochemistry and Immunofluorescence Analysis

For immunohistochemistry analysis, lung tissues were fixed overnight in buffered 10% formaldehyde, embedded in paraffin, and sectioned at a thickness of 4 mm. The sections were stained with hematoxylin and eosin (H&E) using the Hematoxylin and Eosin Stain Kit (H-3502, Vector Laboratories) according to the manufacturer’s protocol. The sections were stained with antibody against specific targets. Secondary antibody was horseradish peroxidase (HRP)-conjugated anti-mouse (SC-516102, SantaCruz Biotechnology), HRP-conjugated anti-rabbit (SC-2357, SantaCruz Biotechnology), HRP-conjugated anti-rat (SC-2750, SantaCruz Biotechnology), Goat anti-Mouse IgG1 Cross-Adsorbed Secondary Antibody, Alexa Fluor 488 (A-21121, Thermo Fisher Scientific), and Goat anti-Rabbit IgG (H + L) Highly Cross-Adsorbed Secondary Antibody, Alexa Fluor 555 (A-21429, Thermo Fisher Scientific). Chromogenic detection was performed using DAB Peroxidase (HRP) Substrate Kit (SK-4100, Vector Laboratories) according to the manufacturer’s protocol. Lung sections were counterstained with Hematoxylin Solution, Harris Modified (HHS32, Sigma). For lung fibrosis analysis, Masson’s trichrome (M/T) staining was performed using trichrome stain kit (ab150686, Abcam) according to the manufacturer’s instructions. Stained lung sections were analyzed by Olympus BX53M microscope and quantified by using Olympus Stream software and ImageJ software v1.52a (Bethesda, MD, USA). For immunofluorescence analysis, lung sections were stained with antibody against specific targets. Lung sections were analyzed using a Zeiss LSM880 laser scanning confocal microscope.

### 2.9. Cytokine Analysis

Supernatants from alveolar macrophages and BMDMs were measured for mouse IL-1β (MLB00C, R&D systems (Minneapolis, MN, USA)), mouse IL-18 (7625, R&D systems (Minneapolis, MN, USA)), and mouse TNF-α (MTA00B, R&D systems (Minneapolis, MN, USA)) according to the manufacturer’s instructions.

### 2.10. ASC Speck Formation Assay

The WT alveolar macrophages and BMDMs were seeded on chamber slides. After LPS and poly(dA:dT) stimulation, cells were fixed with 4% paraformaldehyde and then incubated with polyclonal ASC antibody (ADI-905-173-100, Enzo Lifesciences) for 16 h and FITC goat anti-rabbit (IgG) secondary antibody (ab6717, Abcam (Cambridge, MA, USA)) for 1h followed by DAPI (P36962, ThermoFisher Scientific) staining [41,42,43]. The ASC specks were analyzed using a Zeiss LSM880 laser scanning confocal microscope and quantified using ImageJ software v1.52a (Bethesda, MD, USA). The graph in figure represents the quantification of percent of ASC speck-positive cells for each mouse.

### 2.11. Statistical Analysis

All data are mean ± SD, combined from three independent experiments. All statistical tests were analyzed using a two-tailed Student’s *t*-test for comparison of two groups, and analysis of variance (ANOVA) (with post hoc comparisons using Dunnett’s test), using a statistical software package (GraphPad Prism version 4.0, GraphPad Software Inc. (San Diego, CA, USA)) for comparison of multiple groups.

## 3. Results

### 3.1. The DROSHA and AIM2 Protein Levels were Elevated in Lung Tissues of Patients with IPF

To investigate the role of DROSHA during lung fibrosis in patients with IPF, we analyzed whether the DROSHA protein levels were elevated in lung tissues from patients with IPF (Table 1). We first measured lung fibrosis by Masson’s trichrome staining (M/T) in lung tissues from patients with IPF (IPF) or non-IPF patients (Control) (Figure 1A). Lung fibrosis is highly increased in patients with IPF (IPF) relative to non-IPF patients (Control) (Figure 1A). We next measured the DROSHA protein levels in lung tissues from patients with IPF (IPF) or non-IPF patients (Control) using immunohistochemistry staining (Figure 1B). Notably, immunohistochemistry staining revealed the DROSHA protein expression was significantly elevated in alveolar macrophages of patients with IPF (IPF) compared to non-IPF patients (Control) (Figure 1B). In contrast to DROSHA, the DGCR8 protein expression was comparable (Appendix A). Consistent with immunohistochemistry staining, the DROSHA protein levels were significantly increased in lung tissues from patients with IPF (IPF) compared to non-IPF patients (Control) (Figure 1C), whereas the DGCR8 protein levels were unchanged (Figure 1C). Next, we investigated whether the AIM2 inflammasome was increased in lung tissues from patients with IPF. We measured the AIM2 protein levels in lung tissues from patients with IPF. Similar to DROSHA expression, the AIM2 protein levels were significantly increased in lung tissues from patients with IPF (IPF) compared to non-IPF patients (Control) (Figure 1D). These results suggest that the DROSHA and AIM2 protein levels were elevated in patients with IPF.

### 3.2. The DROSHA and AIM2 Expression Levels were Elevated in Alveolar Macrophages of Patients with IPF

Next, we investigated the role of DROSHA in AIM2 inflammasome-dependent lung inflammation during IPF. We first analyzed whether the DROSHA expression levels were increased in alveolar macrophages of patients with IPF using immunofluorescence staining. We measured the changes of DROSHA and AIM2 expression levels in cluster of differentiation 68 (CD68)-positive alveolar macrophages [44] of patients with IPF (IPF) and non-IPF patients (Control). Immunofluorescence staining revealed that the intensity and number of DROSHA-positive staining in CD68-positive alveolar macrophages were increased in patients with IPF (IPF) relative to non-IPF patients (Control) (Figure 2A). Next, we investigated whether the AIM2 expression levels were elevated in CD68-positive alveolar macrophages of patients with IPF using immunofluorescence staining (Figure 2B). Consistent with DROSHA expression, the intensity and number of AIM2-positive staining in CD68-positive alveolar macrophages were increased in patients with IPF (IPF) compared to non-IPF patients (Control) (Figure 2B). Moreover, we examined whether the positive subcellular co-localization between DROSHA and AIM2 is elevated in patients with IPF. DROSHA is co-localized with AIM2 in patients with IPF (IPF) (Figure 2C). Notably, the intensity and number of cells which have the positive subcellular co-localization between DROSHA and AIM2 were significantly increased in patients with IPF (IPF) relative to non-IPF patients (Control) (Figure 2C). These results suggest that the DROSHA and AIM2 expression levels in alveolar macrophages were elevated in patients with IPF.

### 3.3. The DROSHA Protein Levels were Elevated in Alveolar Macrophages during Bleomycin-Induced Pulmonary Fibrosis

We investigated the role of DROSHA in alveolar macrophages in a mouse model of bleomycin-induced pulmonary fibrosis. We analyzed whether the DROSHA expression levels were elevated in alveolar macrophages of lung tissues from a mouse model of bleomycin-induced pulmonary fibrosis using immunohistochemistry staining. Notably, the density and number of DROSHA-positive staining in alveolar macrophages were increased in mice treated with bleomycin (Bleomycin) compared to mice treated with vehicle control (PBS) (Figure 3A). In contrast, the DGCR8 expression levels were comparable (Appendix A). Consistently, the DROSHA protein levels were elevated in lung tissues of mice treated with bleomycin (Bleomycin) relative to mice treated with vehicle control (PBS) (Figure 3B). In contrast, the DGCR8 protein levels were unchanged (Figure 3B). Moreover, we examined whether the DROSHA expression levels in alveolar macrophages were increased in mice treated with bleomycin using immunofluorescence staining. We analyzed the DROSHA expression levels in F4/80-positive alveolar macrophages [45] of mice treated with bleomycin (Bleomycin) or vehicle control (PBS). The intensity and number of DROSHA-positive staining in F4/80-positive alveolar macrophages were increased in lung tissues of mice treated with bleomycin (Bleomycin) relative to mice treated with vehicle control (PBS) (Figure 3C). These results suggest that the DROSHA protein levels in alveolar macrophages were elevated in bleomycin-induced pulmonary fibrosis.

### 3.4. Deficiency of DROSHA Suppresses the AIM2 Inflammasome Activation in Alveolar Macrophages

Since DROSHA and AIM2 expression levels were elevated in alveolar macrophages during lung fibrosis, we next investigated the function of DROSHA during AIM2 inflammasome activation in alveolar macrophages. We analyzed whether the deficiency of DROSHA could suppress the secretion of IL-1β and IL-18 in lipopolysaccharide (LPS)-primed WT alveolar macrophages in response to poly(dA:dT), an AIM2 inflammasome activator. We used DROSHA-targeted guide RNA (Drosha gRNA) to delete mouse DROSHA in primary mouse alveolar macrophages via clustered regularly interspaced short palindromic repeats (CRISPR) technology (Figure 4A). The DROSHA-targeted gRNA suppressed the secretion of IL-1β and IL-18 compared to control plasmid (control), whereas the secretion of tumor necrosis factor (TNF)-α was unchanged (Figure 4A). Consistently, the DROSHA-targeted gRNA inhibited the activation of caspase-1 and IL-1β cleavage relative to control plasmid (Figure 4B). In contrast, the DROSHA-targeted gRNA did not change on the secretion of IL-1β and IL-18 in response to ATP (a NLRP3 inflammasome activator), flagellin (an NLRC4 inflammasome activator), or muramyldipeptide (MDP) (an NLRP1 inflammasome activator) (Figure 4C). Moreover, we investigated whether DROSHA-targeted gRNA could suppress the formation of ASC specks, which is required for AIM2 inflammasome activation [41,42]. Notably, the DROSHA-targeted gRNA significantly reduced the formation of ASC specks by LPS and poly(dA:dT) stimulation compared to the control plasmid (Figure 4D). In contrast to alveolar macrophages, LPS and poly(dA:dT) stimulation did not changes the activation of primary mouse lung fibroblasts (Appendix A). These results suggest that deficiency of DROSHA suppresses the AIM2 inflammasome activation in alveolar macrophages.

### 3.5. Deficiency of DROSHA Suppresses the AIM2 Inflammasome Activation in Macrophages

We next examined whether the deficiency of DROSHA can suppress the AIM2 inflammasome activation in bone marrow-derived macrophages (BMDMs). We used DROSHA-targeted small interfering RNA (*Drosha* siRNA) to delete mouse DROSHA in primary mouse BMDMs (Figure 5A). Consistent with deficiency of DROSHA in alveolar macrophages (Figure 4), the *Drosha* siRNA significantly suppressed the secretion of IL-1β and IL-18 relative to control siRNA (Control siRNA), whereas the secretion of TNF-α was unchanged (Figure 5A). Moreover, the *Drosha* siRNA reduced the activation of caspase-1 and IL-1β cleavage relative to control siRNA (Figure 5B). In contrast, the *Drosha* siRNA did not change on the secretion of IL-1β and IL-18 in response to ATP, flagellin, or MDP compared to control siRNA (Figure 5C). Furthermore, the *Drosha*-siRNA significantly inhibited the formation of ASC specks by LPS and poly(dA:dT) stimulation relative to the control siRNA (Figure 5D). These results suggest that deficiency of DROSHA suppresses the AIM2 inflammasome activation in macrophages.

### 3.6. Transduction of miRNA Promotes the ASC Speck Formation for AIM2 Inflammasome Activation

We next investigated the underlying molecular mechanism by which DROSHA regulates AIM2 inflammasome activation in macrophages. Since DROSHA-induced miRNAs have double-stranded RNA (dsRNA) structure of the hairpins in a pri-miRNA and can bind to dsDNA to form hetero-triplex structures at specific target sequences in DNA [46], we examined whether the transduction of miRNA can directly promote the AIM2 inflammasome activation in response to poly(dA:dT). We transduced the five independent synthetic miRNAs (miRNA #1, #2, #3, #4, and #5) to increase the amount of miRNA during AIM2 inflammasome activation in macrophages. The transduction of miRNAs significantly elevated the secretion of IL-1β and IL-18 compared to vehicle control, whereas the secretion of TNF-α was unchanged (Figure 6A). Consistently, the transduction of miRNAs increased the activation of caspase-1 and IL-1β cleavage relative to vehicle control (Figure 6B). We next analyzed whether the transduction of miRNA could promote the formation of ASC specks for AIM2 inflammasome activation. Notably, the transduction of miRNA enhanced the formation of ASC specks by LPS and poly(dA:dT) stimulation relative to the vehicle control (Figure 6C). These results suggest that the tranduction of miRNA promotes the complex formation of AIM2 inflammasome for AIM2 inflammasome activation.

## 4. Discussion

Here we demonstrate that DROSHA contributes to AIM2 inflammasome activation-dependent lung inflammation during idiopathic pulmonary fibrosis. Our results show that both DROSHA and AIM2 protein levels are significantly elevated in alveolar macrophages of patients with IPF and bleomycin-treated mice. We showed that the genetic inhibition of DROSHA suppresses AIM2 inflammasome-dependent caspase-1 activation and IL-1β and IL-18 secretion by inhibition of ASC speck formation. Furthermore, the transduction of miRNA promotes the AIM2 inflammasome activation. To the best of our knowledge, this is the first reported link between DROSHA-dependent miRNA biosynthesis and AIM2 inflammasome activation in pulmonary fibrosis. Our findings suggest DROSHA-dependent AIM2 inflammasome activation contributes to pulmonary fibrosis.

Previous studies have shown that alveolar macrophages are critical regulators of lung fibrosis [47]. Alveolar macrophages-derived cytokines are associated with the proliferation of fibroblast [48,49]. Also, bleomycin-induced lung fibrosis is linked to the activation of inflammation [50]. Among the various cytokines, IL-1β can directly induce collagen secretion by fibroblasts [48,49]. Transient and chronic expression of IL-1β mediates fibrosis [48,49]. Since inflammasome is a critical regulator of IL-1β production and secretion in alveolar macrophages, we investigated to discover the upstream novel molecule for AIM2 inflammasome activation in alveolar macrophages during IPF. The DROSHA–DGCR8 complex is required for the first step of miRNA biogenesis [51]. DROSHA is a double-strand RNA (dsRNA)-specific endoribonuclease that is involved in the initial step of miRNA biogenesis [40]. Although the role of miRNA has been demonstrated in the immune response [31], the function of DROSHA in AIM2 inflammasome activation has not previously been reported. Our results suggest that DROSHA could be a critical molecule for AIM2 inflammasome activation-dependent IL-1β production in alveolar macrophages during IPF.

The changes in miRNA are important for the immune response such as maturation, proliferation, differentiation, and activation of immune cells [51,52]. The total amount of miRNA was elevated in bronchoalveolar lavage (BAL) from patients with IPF compared to controls [29]. These previous studies indicated that miRNA could have a role as a DAMP in lung inflammation during IPF. Consistent with these previous studies, we demonstrate that excessive accumulation of miRNAs promotes the production of IL-1β and IL-18 via AIM2 inflammasome activation in macrophages. Moreover, we show that the DROSHA levels were elevated in alveolar macrophages in patients with IPF. These results suggest that the DROSHA-dependent miRNA production and the excessive accumulation of miRNA can regulate AIM2 inflammasome activation in alveolar macrophages during IPF. Although we showed the elevated expression of DROSHA in patients with IPF, there is limitation of small cohort number in our human study with patients with IPF. To investigate the role of DROSHA in the pathogenesis of IPF, the correlation between DROSHA expression and the progression of IPF needs to be clarified by further studies.

Although the function of individual miRNA during IPF was reported in the previous papers, the role of miRNA on AIM2 inflammasome activation has not been discovered the mechanism for the progression of IPF. As an upstream molecule of AIM2 inflammasome activation in alveolar macrophages of lung during IPF, we demonstrate that miRNA could be a critical activator for AIM2 inflammasome-mediated lung inflammation. We show that inhibition of DROSHA suppressed AIM2 inflammasome-dependent caspase-1 activation and ASC speck formation in response to poly(dA:dT) as dsDNA which is required for AIM2 inflammasome activation. Consistent with these results, the high levels of miRNA by transduction increased AIM2 inflammasome-dependent caspase-1 activation and AIM2 inflammasome complex formation via ASC speck formation by stimulation with poly(dA:dT). Our results suggest that DROSHA-mediated miRNA production or secretion may be a critical signature for the AIM2 inflammasome activation during IPF.

In summary, we found that (1) DROSHA is elevated in alveolar macrophages in both human IPF and a mouse model of pulmonary fibrosis, and (2) DROSHA promotes AIM2 inflammasome activation. Our results support that DROSHA-driven AIM2 inflammation activation could be a critical molecular pathway in lung inflammation during IPF. Although not yet fully completed, our study provides the role of DROSHA in AIM2 inflammasome-dependent lung inflammation in the pathogenesis of pulmonary fibrosis.

## Figures and Tables

**Figure 1 cells-08-00938-f001:**
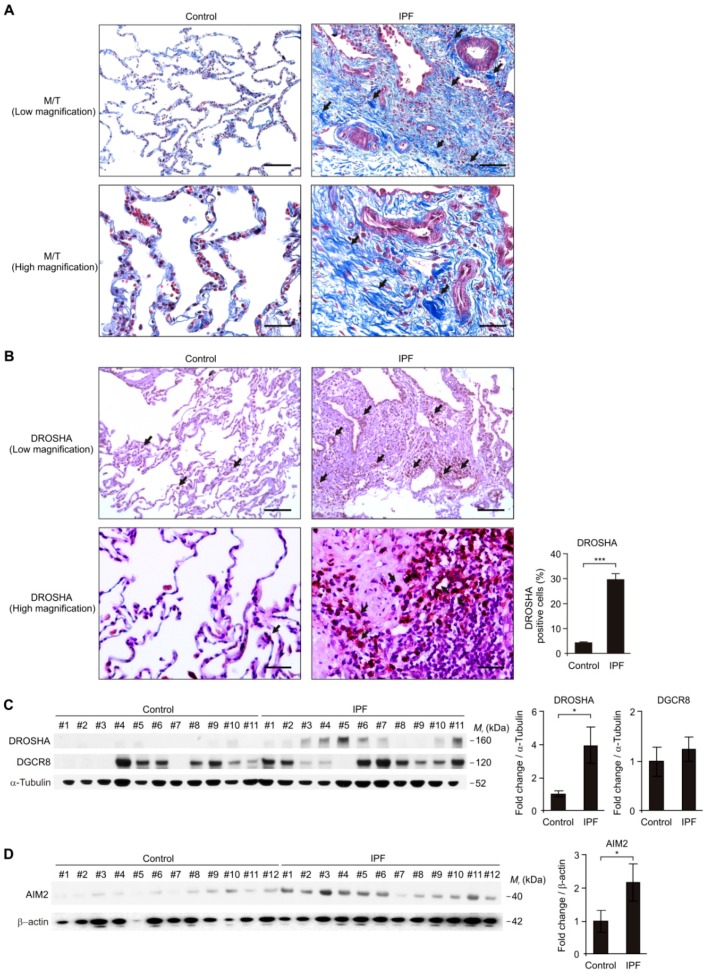
The DROSHA protein levels were elevated in lung tissues of patients with idiopathic pulmonary fibrosis (IPF). (**A**) Representative immunohistochemistry image of fibrosis by Masson’s trichrome (M/T) staining in lung tissues from patients with IPF (IPF) or non-IPF patients (Control). Positive area and cells are indicated by black arrow. In the low magnification image (top), scale bars are 200 μm. In the high magnification image (bottom) scale, bars are 50 μm. Results are representative of three independent experiments. (**B**) Representative immunohistochemistry image (left) of DROSHA and quantification of DROSHA-positive cells (the percent of DROSHA-positive cells in total 100 cells in 10 individual images per group) (right) in lung tissues from patients with IPF (IPF, *n* = 12) or non-IPF patients (Control, *n* = 9). Positive area and cells are indicated by a black arrow. In low magnification image (top), scale bars are 200 μm. In high magnification image (bottom) scale, bars are 50 μm. (**C**) Representative immunoblot analysis for DROSHA and DGCR8 (left) and densitometry quantification of DROSHA and DGCR8 levels (normalized to levels of α-tubulin) (right) from lung tissues from patients with IPF (IPF, *n* = 11) or non-IPF patients (Control, *n* = 11). (**D**) Representative immunoblot analysis for absent in melanoma 2 (AIM2) (left) and densitometry quantification of AIM2 levels (normalized to levels of β-actin) (right) from lung tissues from patients with IPF (IPF, *n* = 12) or non-IPF patients (Control, *n* = 12). For immunoblots, α-tubulin or β-actin was used as loading control. Data are representative of three independent experiments. Data are mean ± SEM. *** *p* <0.001, * *p* <0.05; by Student’s two-tailed *t*-test.

**Figure 2 cells-08-00938-f002:**
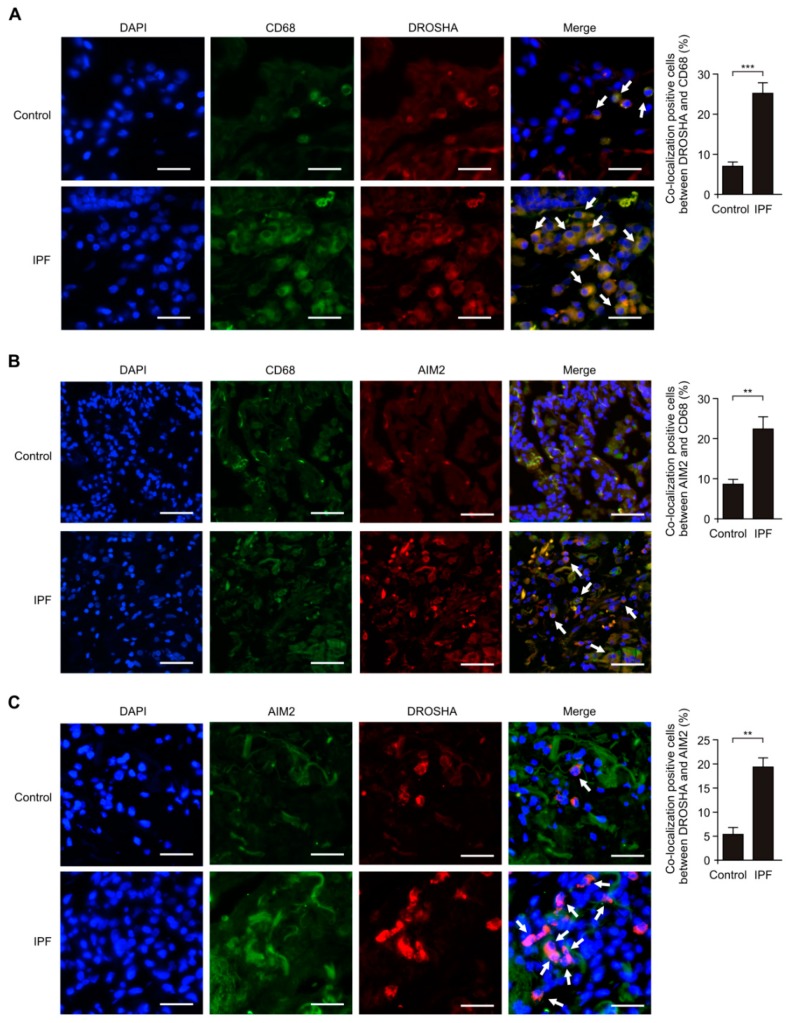
The DROSHA and AIM2 expression levels were elevated in alveolar macrophages of patients with IPF. (**A**) Representative immunofluorescence image of CD68 (Green), DROSHA (Red), and DAPI (Blue) staining in lung tissues from patients from IPF (IPF) or healthy subjects (Control). Positive area and cells are indicated by white arrows. Scale bars, 200 μm. Quantification of co-localization positive cells between DROSHA and CD68 (The percent of co-localization positive cells in total 100 cells in 10 individual images per group) (right) in lung tissues from patients with IPF (IPF, *n* = 5) or non-IPF patients (Control, *n* = 5). (**B**) Representative immunofluorescence image of CD68 (Green), AIM2 (Red) and DAPI (Blue) staining in lung tissues from patients with IPF or non-IPF patients (Control). Positive area and cells are indicated by white arrows. Scale bars, 200 μm. Quantification of co-localization positive cells between AIM2 and CD68 (the percent of co-localization positive cells in total 100 cells in 10 individual images per group) (right) in lung tissues from patients with IPF (IPF, *n* = 5) or non-IPF patients (Control, *n* = 5). (**C**) Representative immunofluorescence image of AIM2 (Green), DROSHA (Red), and DAPI (Blue) staining in lung tissues from patients with IPF or non-IPF patients (Control). Positive area and cells are indicated by white arrows. Scale bars, 200 μm. Quantification of co-localization positive cells between DROSHA and AIM2 (the percent of co-localization positive cells in total 100 cells in 10 individual images per group) (right) in lung tissues from patients with IPF (IPF, *n* = 3) or non-IPF patients (Control, *n* = 3). Data are mean ± SEM. *** *p* <0.001, ** *p* <0.01; by Student’s two-tailed *t*-test.

**Figure 3 cells-08-00938-f003:**
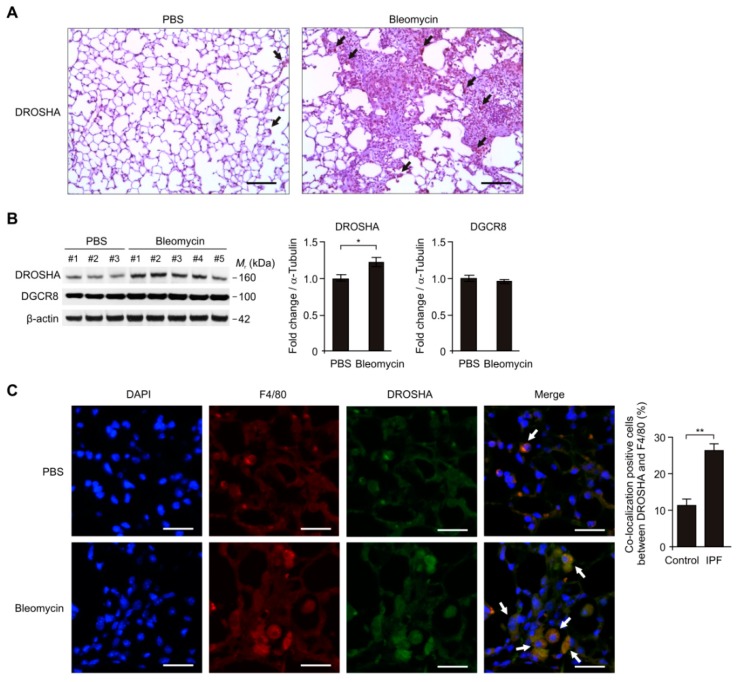
The DROSHA protein levels were elevated in alveolar macrophages during bleomycin-induced pulmonary fibrosis. (**A**) Representative immunohistochemistry image of DROSHA in lung tissues from wild-type (WT) mice were exposed to PBS or bleomycin via oropharyngeal aspiration. Positive area and cells are indicated by the black arrow. Scale bars, 200 μm. Results are representative of three independent experiments. (**B**) Representative immunoblot analysis for DROSHA and DGCR8 (left) and densitometry quantification of DROSHA and DGCR8 levels (normalized to levels of β-actin) (right) in lung tissues from WT mice exposed to PBS (*n* = 3) or bleomycin (*n* = 5) via oropharyngeal aspiration. For immunoblots, β-actin was used as loading control. Data are representative of three independent experiments. Data are mean ± SEM. * *p* <0.05; by Student’s two-tailed *t*-test. (**C**) Representative immunofluorescence image of F4/80 (Red), DROSHA (Green), and DAPI (Blue) staining in lung tissues from WT mice exposed to PBS or bleomycin via oropharyngeal aspiration. Positive area and cells are indicated by white arrows. Scale bars, 200 μm. Quantification of co-localization positive cells between DROSHA and F4/80 (the percent of co-localization positive cells in total 100 cells in 10 individual images per group) (right) in lung tissues from WT mice exposed to bleomycin (*n* = 5) or PBS (*n* = 3) via oropharyngeal aspiration. Data are mean ± SEM. ** *p* <0.01, * *p* <0.05; by Student’s two-tailed *t*-test.

**Figure 4 cells-08-00938-f004:**
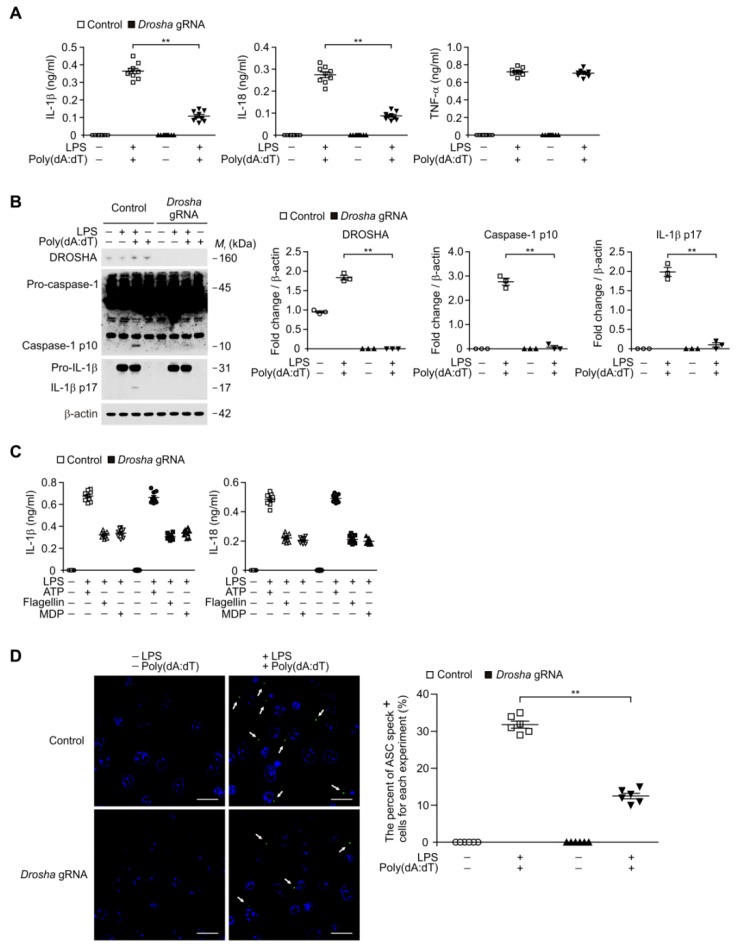
Deficiency of DROSHA suppresses AIM2 inflammasome activation in alveolar macrophages. (**A**) Quantification of interleukin (IL)-1β (left), IL-18 (middle), and TNF-α (right) secretion from wild-type (WT) alveolar macrophages transduced with DROSHA-targeting gRNA (*Drosha* gRNA), or with a control plasmid (Control), and stimulated with lipopolysaccharide (LPS) and poly(dA:dT). (*n* = 9 mice per group). (**B**) Representative immunoblot analysis for caspase-1 and IL-1β (left) and densitometry quantification of caspase-1 p10 and IL-1β p17 levels (normalized to levels of β-actin) (right) from WT alveolar macrophages transduced with DROSHA-targeting gRNA (*Drosha* gRNA), or with a control plasmid (Control), and stimulated with LPS and poly(dA:dT). For immunoblots, β-actin was used as loading control. (*n* = 3 mice per group). (**C**) Quantification of IL-1β and IL-18 secretion from WT alveolar macrophages transduced with DROSHA-targeting gRNA (*Drosha* gRNA), or with a control plasmid (Control), and stimulated with LPS and either ATP, flagellin, or MDP (*n* = 9 mice per group). (**D**) Representative immunofluorescence images (total 100 cells in 10 individual images per group) (left) and quantification (right) of ASC speck formation (white arrows) (the number of ASC speck positive cells in 10 individual images per group) in WT alveolar macrophages transduced with DROSHA-targeting gRNA (*Drosha* gRNA), or with a control plasmid (Control), and stimulated with LPS and poly(dA:dT). (*n* = 6 mice per group). Scale bars, 20 μm. Data are mean ± SEM. ** *p* <0.01; by Student’s two-tailed *t*-test or ANOVA.

**Figure 5 cells-08-00938-f005:**
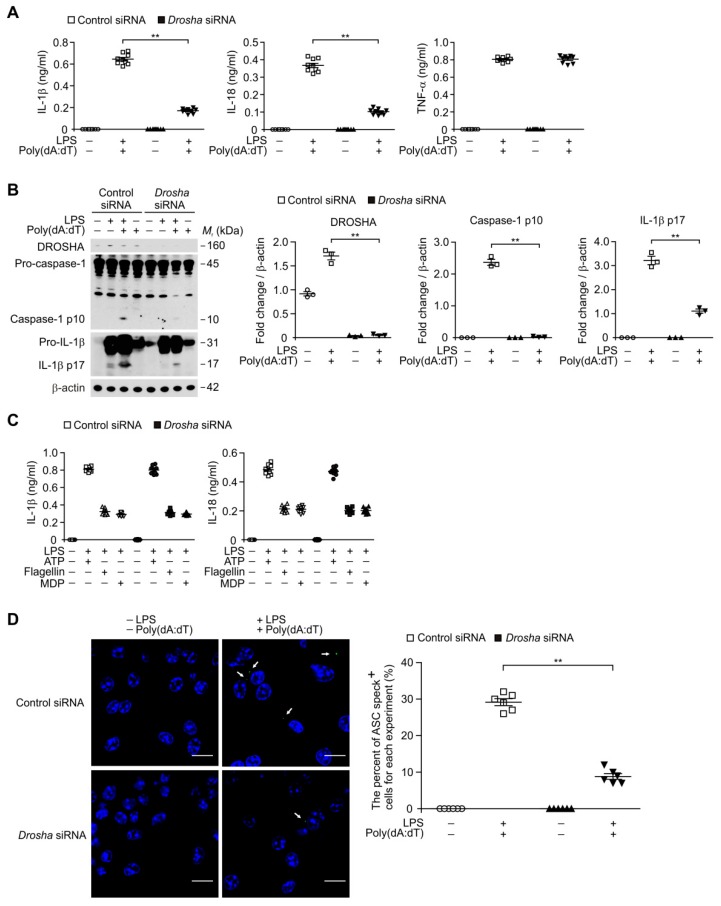
Deficiency of DROSHA suppresses AIM2 inflammasome activation in macrophages. (**A**) Quantification of IL-1β (left), IL-18 (middle) and TNF-α (right) secretion from WT bone marrow-derived macrophages (BMDMs) transduced with DROSHA -targeting small interfering RNA (*Drosha* siRNA), or with a control plasmid (Control), and stimulated with LPS and poly(dA:dT) (*n* = 9 mice per group). (**B**) Representative immunoblot analysis for caspase-1 and IL-1β (left) and densitometry quantification of caspase-1 p10 and IL-1β p17 levels (normalized to levels of β-actin) (right) from WT BMDMs transduced with DROSHA -targeting siRNA (*Drosha* siRNA), or with a control plasmid (Control), and stimulated with LPS and poly(dA:dT). For immunoblots, β-actin was used as loading control (*n* = 3 mice per group). (**C**) Quantification of IL-1β and IL-18 secretion from WT BMDMs transduced with DROSHA -targeting siRNA (*Drosha* siRNA), or with a control plasmid (Control), and stimulated with LPS and either ATP, flagellin, or MDP (*n* = 9 mice per group). (**D**) Representative immunofluorescence images (total 100 cells in 10 individual images per group) (left) and quantification (right) of ASC speck formation (white arrows) (the number of ASC speck positive cells in 10 individual images per group) in WT BMDMs transduced with DROSHA -targeting siRNA (*Drosha* siRNA), or with a control plasmid (Control), and stimulated with LPS and poly(dA:dT). (*n* = 6 mice per group). Scale bars, 20 μm. Data are mean ± SEM. ** *p* <0.01; by Student’s two-tailed *t*-test or ANOVA.

**Figure 6 cells-08-00938-f006:**
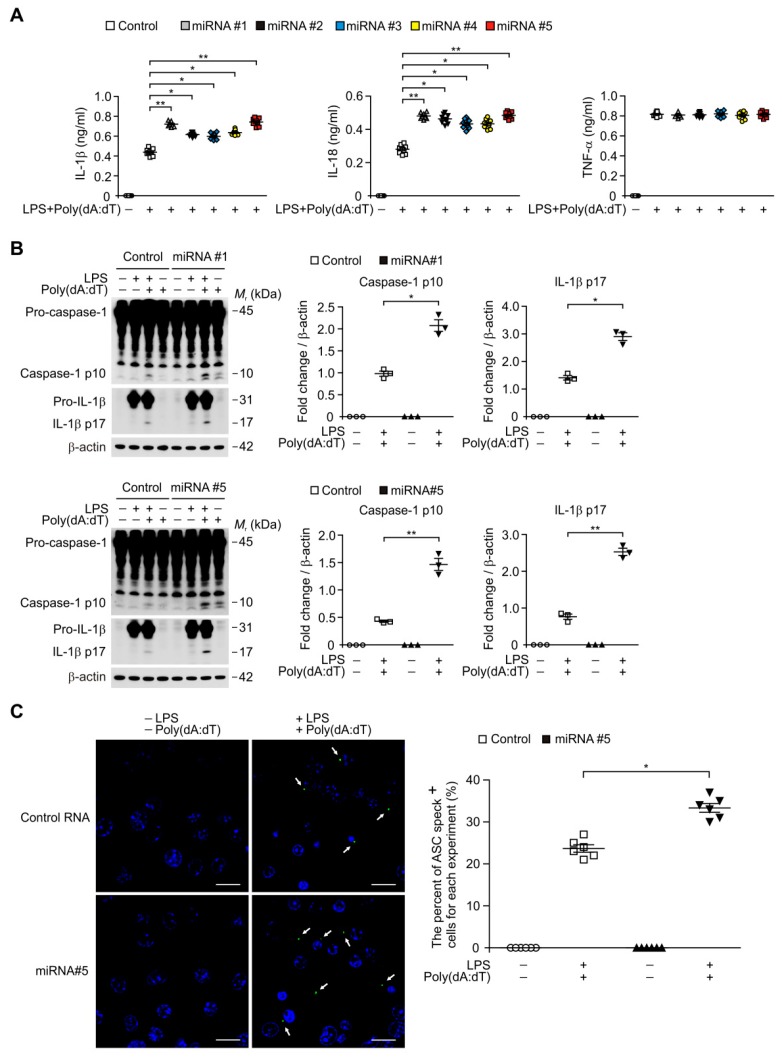
MicroRNA (miRNA) promotes AIM2 inflammasome activation by ASC speck formation in macrophages. (**A**) Quantification of IL-1β (left), IL-18 (middle), and TNF-α (right) secretion from WT BMDMs transduced with five independent miRNA (miRNA#1, miRNA#2, miRNA#3, miRNA#4 and miRNA#5), or with a control RNA (Control), and stimulated with LPS and poly(dA:dT) (*n* = 9 mice per group). (**B**) Representative immunoblot analysis for caspase-1 and IL-1β (left) and densitometry quantification of caspase-1 p10 and IL-1β p17 levels (normalized to levels of β-actin) (right) from WT BMDMs transduced with two independent miRNA (miRNA#1 and miRNA#5), or with a control RNA (Control), and stimulated with LPS and poly(dA:dT). For immunoblots, β-actin was used as loading control. (*n* = 3 mice per group). (**C**) Representative immunofluorescence images (total 100 cells in 10 individual images per group) (left) and quantification (right) of ASC speck formation (white arrows; the number of ASC speck-positive cells in 10 individual images per group) in WT BMDMs transduced with miRNA (miRNA#5), or with a control RNA (Control), and stimulated with LPS and poly(dA:dT) (*n* = 6 mice per group). Scale bars, 20 μm. Data are mean ± SEM. ** *p* <0.01, * *p* <0.05; by Student’s two-tailed *t*-test or ANOVA.

**Table 1 cells-08-00938-t001:** Baseline characteristics of non-IPF control and patients with IPF.

	Control	IPF
Subjects	12	12
Age years	46.5+/−11.4	65+/−9.2
Male/female	11/1	11/1
Smoking status yes/no	4/8	6/6

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
