# Peer review of "DROSHA-Dependent AIM2 Inflammasome Activation Contributes to Lung Inflammation during Idiopathic Pulmonary Fibrosis"

_cells, 2019, doi:10.3390/cells8080938_

Round 1

Reviewer 1 Report

The manuscript by Cho et al. aimed to investigate the role of DROSHA-dependent AIM2 inflammasome activation in the inflammation of idiopathic pulmonary fibrosis (IPF). This is an interesting study; however, there were several deficits needing to be improved to increase the scientific richness and perfectness.

1. The scientific writing should be improved

(1) What is ASC? It should be defined in this study. For reader friendly, authors should provide background information about ASC and the formation of inflammasome in introduction.

(2) Discussion is too brief and should be largely revised. These findings should be discussed in depth.

(3) The style of Ref. title should be revised in Ref. 20, 22 and 38.

(4) It will be nice to add the Graphic abstract in revised manuscript.

2. Authors should experimentally address the comments as listed below:

(1) In figure 1, the expression of AIM2 should be examined. Additionally, DROSHA expression was not changed in > 50% of IPF patients. Does DROSHA play a crucial role in pathogenesis of IPF?

(2) Higher magnification images are required for the histology in figure 1.

(3) In figure 2, images with low magnification are required. From the images of a) and b) panels, the reviewer cannot see the difference in the structure of lung in control and IPF groups.

(4) Similar to the above comment, the figure 3 should be revised.

(5) The link between in vivo study and in vitro model was lost. In cell system, LPS and poly(dA:dT) were used; however, these two factors seem not to be key players in fibrosis.

(6) How about the roles of DROSHA and AIM2 in fibrosis? Do DROSHA and AIM2 expression in fibroblasts of IPF lesions? The model of fibroblasts with TGF-beta, LPS or poly(dA:dT) treatment is highly recommended.

Author Response

Response to Cells Reviewer 1 Comments

The manuscript by Cho et al. aimed to investigate the role of DROSHA-dependent AIM2 inflammasome activation in the inflammation of idiopathic pulmonary fibrosis (IPF). This is an interesting study; however, there were several deficits needing to be improved to increase the scientific richness and perfectness.

The scientific writing should be improved

(1) What is ASC? It should be defined in this study. For reader friendly, authors should provide background information about ASC and the formation of inflammasome in introduction.

Response 1. (1) :

According to the Reviewer’s request, we have provided more detail description that what is ASC and the formation of inflammasome in the introduction.

To reflect this correction, the following text has been added to the Introduction section:

Introduction, page 2 line 53. “Inflammasomes are consist with sensor proteins, such as NOD-, LRR- and pyrin domain-containing 1 (NLRP1), NLRP3, NLRC4 or PYHIN (pyrin and HIN domain-containing protein) family members absent in melanoma 2 (AIM2), caspase-1 and the adaptor apoptosis-associated speck-like protein containing a caspase recruitment domain (ASC), which is an adaptor protein encoded by PYCARD that is a common protein to all inflammasomes [12-14]. In the formation of inflammasome, ASC interacts with both the upstream inflammasome sensor protein and caspase-1 and this interaction triggers the assembly of ASC as a large protein speck for inflammasome activation [15, 16].”

Introduction, page 2 line 63. “Binding of dsDNA to AIM2 leads to the activation of AIM2 inflammasome complex by recruitment of ASC via PYD-PYD interactions and caspase-1 via CARD-CARD interactions [19]. The formation of AIM2 inflammasome promotes the increase of caspase-1 p10 fragments as an active form. Activated caspase-1 p10 fragments proteolytically produce activated IL-1β and IL-18 form from pro-IL-1β and pro-IL-18 [20, 21].”

(2) Discussion is too brief and should be largely revised. These findings should be discussed in depth.

Response 1. (2) :

According to the Reviewer’s request, we provided the expanded discussion for our finding related to DROSHA and AIM2 inflammasome.

The following text has been added to the Discussion section:

Discussion, page 17 line 447. Previous studies have shown that alveolar macrophages are critical regulators of lung fibrosis [46]. Alveolar macrophages-derived cytokines are associated with the proliferation of fibroblast [47,48]. Also, bleomycin-induced lung fibrosis is linked to the activation of inflammation [49]. Among the various cytokines, IL-1β can directly induce collagen secretion by fibroblasts [47,48]. Transient and chronic expression of IL-1β mediates fibrosis [47,48]. Since inflammasome is a critical regulator of IL-1β production and secretion in alveolar macrophages, we investigated to discover the upstream novel molecule for AIM2 inflammasome activation in alveolar macrophages during IPF.

Discussion, page 17 line 464.  “Consistent with these previous studies, we demonstrate that excessive accumulation of miRNAs promotes the production of IL-1β and IL-18 via AIM2 inflammasome activation in macrophages. Moreover, we show that the DROSHA levels were elevated in alveolar macrophages in patients with IPF. These results suggest that the DROSHA-dependent miRNA production and the excessive accumulation of miRNA can regulate AIM2 inflammasome activation in alveolar macrophages during IPF. Although we showed the elevated expression of DROSHA in patients with IPF, there is limitation of small cohort number in our human study with patients with IPF. To investigate the role of DROSHA in the pathogenesis of IPF, the correlation between DROSHA expression and the progression of IPF needs to be clarified by further studies.

Although the function of individual miRNA during IPF was reported in the previous papers, the role of miRNA on AIM2 inflammasome activation has not been discovered the mechanism for the progression of IPF. As an upstream molecule of AIM2 inflammasome activation in alveolar macrophages of lung during IPF, we demonstrate that miRNA could be a critical activator for AIM2 inflammasome-mediated lung inflammation. We show that inhibition of DROSHA suppressed AIM2 inflammasome-dependent caspase-1 activation and ASC speck formation in response to poly(dA:dT) as dsDNA which is required for AIM2 inflammasome activation.”

(3) The style of Ref. title should be revised in Ref. 20, 22 and 38.

Response 1. (3) :

According to the Reviewer’s request, we revised the style of Ref. 20, 22 and 38.

To reflect this correction, the following text has been revised in the References section:

References, page 19 line 530.  “27.       Tahamtan A, Teymoori-Rad M, Nakstad B, Salimi V. Anti-inflammatory microRNAs and their potential for inflammatory diseases treatment. Front Immunol 2018, 9, 1377.

Fan L, Yu X, Huang Z, Zheng S, Zhou Y, Lv H, Zeng Y, Xu JF, Zhu X, Yi X. Analysis of microarray-identified genes and microRNAs associated with idiopathic pulmonary fibrosis. Mediators Inflamm 2017, 1804240. “

References, page 20 line 581.  “45.       Paugh SW, Coss DR, Bao J, Laudermilk LT, Grace CR, Ferreira AM, Waddell MB, Ridout G, Naeve D, Leuze M, LoCascio PF, Panetta JC, Wilkinson MR, Pui CH, Naeve CW, Uberbacher EC, Bonten EJ, Evans WE. MicroRNAs form triplexes with double stranded DNA at sequence-specific binding sites; a eukaryotic mechanism via which microRNAs could directly alter gene expression. PLoS Comput Biol. 2016, 12, e1004744. “

(4) It will be nice to add the Graphic abstract in revised manuscript.

Response 1. (4) :

According to the Reviewer’s request, we provided the Graphic abstract for revised manuscript.

Authors should experimentally address the comments as listed below:

(1) In figure 1, the expression of AIM2 should be examined. Additionally, DROSHA expression was not changed in > 50% of IPF patients. Does DROSHA play a crucial role in pathogenesis of IPF?

Response 2. (1) :

According to the Reviewer’s request, we performed additional experiments for the expression of AIM2 protein in lung tissues from patients with IPF or control (see new Fig. 1C). The AIM2 protein levels were significantly increased in lung tissues from patients with IPF (see new Fig. 1C).

The following text has been added to the Results section:

Results, page 6 line 242.  “Next, we investigated whether the AIM2 inflammasome was increased in lung tissues from patients with IPF. We measured the AIM2 protein levels in lung tissues from patients with IPF. Similar to DROSHA expression, the AIM2 protein levels were significantly increased in lung tissues from patients with IPF (IPF) compared to non-IPF patients (Control) (Figure 1D). “

According to the Reviewer’s point, we agree the limitation of small cohort number of IPF patients in our human study. To investigate the role of DROSHA in the pathogenesis of IPF, the correlation between DROSHA expression and the progression of IPF needs to be clarified by further studies. We provided this description in the expanded discussion section.

The following text has been added to the Discussion section:

Discussion, page 17 line 469.  “Although we showed the elevated expression of DROSHA in patients with IPF, there is limitation of small cohort number in our human study with patients with IPF. To investigate the role of DROSHA in the pathogenesis of IPF, the correlation between DROSHA expression and the progression of IPF needs to be clarified by further studies.”

(2) Higher magnification images are required for the histology in figure 1.

Response 2. (2) :

According to the Reviewer’s request, we additionally provided the higher magnification images for the histology in figure 1A and 1B (see new Fig. 1A and 1B).

The following text has been added to the Figure legends:

Figure 1 legend, page 9 line 338. In low magnification image (Top), scale bars are 200 μm. In high magnification image (Bottom) scale, bars are 50 μm.

(3) In figure 2, images with low magnification are required. From the images of a) and b) panels, the reviewer cannot see the difference in the structure of lung in control and IPF groups.

Response 2. (3) :

According to the Reviewer’s request, we provided the low magnification images for figure 2A and 2B (see new Fig. 2A and 2B).

(4) Similar to the above comment, the figure 3 should be revised.

Response 2. (4) :

According to the Reviewer’s request, we provided the low magnification images for figure 3C (see new Fig. 3C).

(5) The link between in vivo study and in vitro model was lost. In cell system, LPS and poly(dA:dT) were used; however, these two factors seem not to be key players in fibrosis.

Response 2. (5) :

According to the Reviewer’s point, we provided the description for the interaction between in vivo study and in vitro model in our study in discussion section. Since alveolar macrophages are critical regulators of lung fibrosis, we tried to discover the upstream novel molecule for AIM2 inflammasome activation in alveolar macrophages during IPF. We used LPS and poly(dA:dT) as an activator for  AIM2 inflammsome activation in alveolar macrophages.

The following text has been added to the Discussion section:

Discussion, page 17 line 447.  “Previous studies have shown that alveolar macrophages are critical regulators of lung fibrosis [46]. Alveolar macrophages-derived cytokines are associated with the proliferation of fibroblast [47,48]. Also, bleomycin-induced lung fibrosis is linked to the activation of inflammation [49]. Among the various cytokines, IL-1β can directly induce collagen secretion by fibroblasts [47,48]. Transient and chronic expression of IL-1β mediates fibrosis [47,48]. Since inflammasome is a critical regulator of IL-1β production and secretion in alveolar macrophages, we investigated to discover the upstream novel molecule for AIM2 inflammasome activation in alveolar macrophages during IPF. The DROSHA-DGCR8 complex is required for the first step of miRNA biogenesis [40]. DROSHA is a double strand RNA (dsRNA)-specific endoribonuclease that is involved in the initial step of miRNA biogenesis [40]. Although the role of miRNA has been demonstrated in the immune response [31], the function of DROSHA in AIM2 inflammasome activation has not previously been reported. Our results suggest that DROSHA could be a critical molecule for AIM2 inflammasome activation-dependent IL-1β production in alveolar macrophages during IPF. “

(6) How about the roles of DROSHA and AIM2 in fibrosis? Do DROSHA and AIM2 expression in fibroblasts of IPF lesions? The model of fibroblasts with TGF-beta, LPS or poly(dA:dT) treatment is highly recommended.

Response 2. (6) :

Since the AIM2 inflammasome components including AIM2, caspase-1 and ASC are only expressed in immune cells such as macrophages not in fibroblast, we analyzed the AIM2 and DROSHA expression in alveolar macrophages of lung tissues in patients with IPF in our study. For the roles of DROSHA and AIM2 in fibrosis, our results suggest that DROSHA-dependent AIM2 inflammsome activation-mediated IL-1β production contributes to lung fibrosis during IPF.

According to the Reviewer’s recommendation, we performed additional experiments whether LPS or poly(dA:dT) treatment could change the activation of fibroblast. We measured the protein expression of collagen-1 and fibronectin in primary mouse lung fibroblasts stimulated with TGF-beta. Although TGF-beta increased the collagen-1 and fibronectin protein levels, LPS and poly(dA:dT), LPS only or poly(dA:dT) only treatment did not change the collagen-1 and fibronectin protein levels. These results indicated that the stimulation with LPS and poly(dA:dT) did not change the activation of fibroblast.

The following text has been added to the Results section:

Results, page 7 line 300.  “In contrast to alveolar macrophages, LPS and poly(dA:dT) stimulation did not changes the activation of primary mouse lung fibroblasts (Supplemental Figure S3). “

Reviewer 2 Report

Major comments 1. DROSHA expression in IPF issues Fig.1: Regarding to DROSHA expression, authors showed representative images, including IHC staining. The key concern is limitation of cohort number, how many samples have authors detected in this experiment, and statistical analysis is necessary. Based on these, related conclusion could be solid. In addition, what about gene expression level of DROSHA in IPF samples, compared to healthy control? 2. Fig.2 Whether subcellular co-localization between DROSHA and AIM2 exists, regarding to this, have authors ever considered or performed related experiments? In addition, related statistical analysis is necessary regarding to immunofluorescence staining images. 3. Fig.3C Results are not convincing, no significant difference between control and bleomycin group. Please re-do statistical analysis. 4. Fig.4 The question is whether LPS stimulates DROSHA expression, please consider it. 5. Western blot of cleaved caspase-1 expression Undoubtedly cleaved caspase-1 expression is the key result in this manuscript. Regarding to cleaved caspase-1 (p10) band, generally, the representative bands in Fig.4,5,6 are totally too weak, so the corresponding conclusions are not solid. Please re-do it through optimizing exposure or use antibody cleaved caspase-1 antibody. Minor comments 1. Human studies Please provide approval number for human studies. 2. Animal experiments Please provide approval number issued by the Institutional Animal Care and Use Committee (IACUC) or other related administrative committees.

Author Response

Response to Cells Reviewer 2 Comments

Major comments

DROSHA expression in IPF issues Fig.1: Regarding to DROSHA expression, authors showed representative images, including IHC staining. The key concern is limitation of cohort number, how many samples have authors detected in this experiment, and statistical analysis is necessary. Based on these, related conclusion could be solid. In addition, what about gene expression level of DROSHA in IPF samples, compared to healthy control?

Response 1 :

According to the Reviewer’s request, we performed additional experiments for statistical analysis of DROSHA expression in patients with IPF (n = 12) or control (n = 9) (see new Fig. 1B).

The following text has been added to the Results section:

Results, page 6 line 241.  “Notably, immunohistochemistry staining revealed the DROSHA protein expression was significantly elevated in alveolar macrophages of patients with IPF (IPF) compared to non-IPF patients (Control) (Figure 1B).”

The following text has been added to the Figure legends section:

Results, page 9 line 346.  “quantification of DROSHA positive cells (The percent of DROSHA positive cells in total 100 cells in 10 individual images per group) (right) in lung tissues from patients with IPF (IPF, n = 12) or healthy subjects (Control, n = 9). ”

Thank you for your suggestion. However, we only have protein lysates for lung tissues of patients with IPF and healthy control in our Human tissues bank. Now, it is hard to get RNA from lung tissues of patients with IPF and healthy control for the gene expression analysis. Analysis for the gene expression level of DROSHA in patients with IPF needs to be clarified by further studies. Please consider.

Fig.2 Whether subcellular co-localization between DROSHA and AIM2 exists, regarding to this, have authors ever considered or performed related experiments? In addition, related statistical analysis is necessary regarding to immunofluorescence staining images.

Response 2 :

Thank you for your suggestion. However, there is time limitation to get new antibody of DROSHA and AIM2 for the co-localization analysis. Now, it is hard to get new antibody within 10 days. Analysis for the co-localization needs to be clarified by further studies. Please consider.

According to the Reviewer’s request, we performed additional experiments for statistical analysis of co-localization between DROSHA and CD68 expression in patients with IPF (n = 5) or control (n = 5) (see new Fig. 2A). And, we also performed additional experiments for statistical analysis of co-localization between AIM2 and CD68 expression in patients with IPF (n = 5) or control (n = 5) (see new Fig. 2B).

The following text has been added to the Results section:

Results, page 6 line 241.  “Notably, immunohistochemistry staining revealed the DROSHA protein expression was significantly elevated in alveolar macrophages of patients with IPF (IPF) compared to non-IPF patients (Control) (Figure 1B).”

The following text has been added to the Figure legends section:

Results, page 9 line 346.  “quantification of DROSHA positive cells (The percent of DROSHA positive cells in total 100 cells in 10 individual images per group) (right) in lung tissues from patients with IPF (IPF, n = 12) or healthy subjects (Control, n = 9). ”

Fig.3C Results are not convincing, no significant difference between control and bleomycin group. Please re-do statistical analysis.

Response 3 :

According to the Reviewer’s request, we performed additional experiments for statistical analysis of the number of positive subcellular co-localization between F4/80 and DROSHA (see new Fig. 3C).

The following text has been added to the Results section:

Results, page 17 line 452.  “Representative immunofluorescence images (total 100 cells in 10 individual images per group) (left) and quantification (right) of positive subcellular co-localization between F4/80 and DROSHA (the percent of cells which have positive subcellular co-localization between F4/80 and DROSHA cells).

Fig.4 The question is whether LPS stimulates DROSHA expression, please consider it. 5. Western blot of cleaved caspase-1 expression Undoubtedly cleaved caspase-1 expression is the key result in this manuscript. Regarding to cleaved caspase-1 (p10) band, generally, the representative bands in Fig.4,5,6 are totally too weak, so the corresponding conclusions are not solid. Please re-do it through optimizing exposure or use antibody cleaved caspase-1 antibody.

Response 4 :

According to the Reviewer’s request, we provided long exposure images for cleaved caspase-1 p10 expression (see new Fig. 4B, 5B, 6B).

Minor comments 1. Human studies Please provide approval number for human studies. 2. Animal experiments Please provide approval number issued by the Institutional Animal Care and Use Committee (IACUC) or other related administrative committees.

Response for minor comment 1 :

According to the Reviewer’s request, we provided approval number for human studies and Animal experiments in Materials and Methods.

The following text has been added to the Materials and Methods section:

Materials and Methods, page 3 line 92.  “Human subjects study was conducted in accordance with the Helsinki Declaration. The Brigham and Women’s Hospital and Weill Cornell Medicine institutional review board approved all experimental procedures involving use of human samples (protocol #: 1405015116A003). “

Materials and Methods, page 3 line 99.  “All mouse experimental protocols were approved by the Institutional Animal Care and Use Committee of Soonchunhyang University (protocol #: SCH18-0032) and Weill Cornell Medicine (protocol #: 2013-0108). “

Round 2

Reviewer 1 Report

Authors have addressed my comments.

Author Response

Response to Cells Reviewer 1 Comments

Authors have addressed my comments.

Response

I appreciate all your comments to improve our paper. Thank you so much.

Reviewer 2 Report

Authors improved quality of manuscript.Considering the importance of co-localization of DROSHA and AIM2, which helps understanding interaction between them, so related experiments are necessary.If possible, please provide related evidence. 

Author Response

Response to Cells Reviewer 2 Comments

Minor comments

Authors improved quality of manuscript. Considering the importance of co-localization of DROSHA and AIM2, which helps understanding interaction between them, so related experiments are necessary. If possible, please provide related evidence.

Response :

I appreciate all your comments to improve our paper.

According to the Reviewer’s request, we performed additional experiments for co-localization of DROSHA and AIM2 in patients with IPF or control (see new Fig. 2C). We also performed statistical analysis of the number of positive subcellular co-localization between DROSHA and AIM2 in patients with IPF (n = 3) or control (n = 3) (see new Fig. 2C).

The following text has been added to the Results section:

Results, page 6 line 269.  “Moreover, we examined whether the positive subcellular co-localization between DROSHA and AIM2 is elevated in patients with IPF. DROSHA is co-localized with AIM2 in patients with IPF (IPF) (Figure 2C). Notably, the intensity and number of cells which have the positive subcellular co-localization between DROSHA and AIM2 were significantly increased in patients with IPF (IPF) relative to non-IPF patients (Control) (Figure 2C).”

The following text has been added to the Figure legends section:

Figure legends, page 10 line 379.  “(C) Representative immunofluorescence image of AIM2 (Green), DROSHA (Red) and DAPI (Blue) staining in lung tissues from patients with IPF or non-IPF patients (Control). Positive area and cells are indicated by white arrows. Scale bars, 200 μm. Quantification of co-localization positive cells between DROSHA and AIM2 (The percent of co-localization positive cells in total 100 cells in 10 individual images per group) (right) in lung tissues from patients with IPF (IPF, n = 3) or non-IPF patients (Control, n = 3).”
